# Effects of Probiotic-Fermented Feed on the Growth Profile, Immune Functions, and Intestinal Microbiota of Bamei Piglets

**DOI:** 10.3390/ani14040647

**Published:** 2024-02-17

**Authors:** Miao Zhang, Zhenyu Yang, Guofang Wu, Fafang Xu, Jianbo Zhang, Xuan Luo, Yuhong Ma, Huili Pang, Yaoke Duan, Jun Chen, Yimin Cai, Lei Wang, Zhongfang Tan

**Affiliations:** 1School of Agricultural Sciences, Zhengzhou University, Zhengzhou 450001, China; miaozhang@zzu.edu.cn (M.Z.); yzy3501670233@163.com (Z.Y.); pang@zzu.edu.cn (H.P.); duyk@nwafu.edu.cn (Y.D.); chenjun6377@163.com (J.C.); cai@affrc.go.jp (Y.C.); 2Plateau Livestock Genetic Resources Protection and Innovative Utilization Key Laboratory of Qinghai Province, Key Laboratory of Animal Genetics and Breeding on Tibetan Plateau, Ministry of Agriculture and Rural Affairs, Qinghai Academy of Animal and Veterinary Medicine, Qinghai University, Xining 810016, China; jim963252@163.com (G.W.); zhangjb9122@163.com (J.Z.); cauluoxuan@163.com (X.L.); qhmkymyh@163.com (Y.M.); 3Bamei Pig Original Breeding Base of Huzhu County, Haidong 810600, China; fafangxu@163.com; 4Japan International Research Center for Agricultural Sciences, Crop, Livestock and Environment Division, Tsukuba 305-8686, Japan

**Keywords:** fermented feed, growth performance, serum immunity, intestinal microbiota, Bamei piglet

## Abstract

**Simple Summary:**

Fermented feed as an alternative to antibiotics in livestock is gaining interest. The purpose of this investigation was to evaluate the effects of the addition of three fermented feeds to the basal diet of Bamei piglets from the Qinghai–Tibetan Plateau. The results suggest that fermented feed has the function of improving the growth condition, immunity and intestinal flora structure of Bamei piglets. This study provides a new approach for using fermented feed as an in-feed antibiotic substitute in the raising of Bamei piglets in high-altitude environments.

**Abstract:**

Purebred Bamei piglets present problems, including slow growth, respiratory disease, and post-weaning stress. This study investigated the effects of *Lactobacillus plantarum* QP28-1- and *Bacillus subtilis* QB8-fermented feed supplementation on the growth performance, immunity, and intestinal microflora of Bamei piglets from Qinghai, China. A total of 48 purebred Bamei piglets (25 days; 6.8 ± 0.97 kg) were divided into the following four groups for a 28-day diet experiment: basal feed (CK); diet containing 10% *Lactobacillus plantarum*-fermented feed (L); diet containing 10% *Bacillus subtilis*-fermented feed (B); and diet containing a mixture of 5% *Lactobacillus plantarum* + 5% *Bacillus subtilis*-fermented feed (H). The daily weight gain and daily food intake of group H increased (*p* < 0.05), and the feed/weight gain ratios of the groups fed with fermented feed decreased more than that of the CK group. The levels of three immune factors, namely immunoglobulin (Ig)M, IgG, and interferon-γ, were higher (*p* < 0.05), whereas those of tumor necrosis factor-α, interleukin (IL)-1β, and IL-6 were lower (*p* < 0.05) in the fermented feed groups than in the CK group. Total protein was higher (*p* < 0.05), while urea nitrogen, total cholesterol and triglycerides were lower (*p* < 0.05) in the mixed-fermented feed group than in the CK group. Analysis of the gut microbiota showed that the addition of fermented feed increased the α-diversity of the gut microbiota, increasing the abundances of probiotics including *Lactobacillus*, *Muribaculaceae*, *Ruminococcaceae*, *Prevotellaceae*, and *Rikenellaceae*. Additionally, correlation analysis demonstrated that several of these probiotic bacteria were closely related to serum immunity. In conclusion, fermented feed supplementation rebuilt the intestinal microbiota of Bamei piglets, thereby reducing the feed/weight ratio, improving feed intake, and enhancing immunity.

## 1. Introduction

Post-weaning piglets are often susceptible to stress when faced with changes in dietary and nutritional structure, psychology, and environment, often resulting in reduced feed intake, dysbiosis of the gut flora, impaired immune function, susceptibility to infection, and slow growth [1,2,3]. Antibiotics are often used for the antimicrobial treatment of piglets; however, antibiotics can negatively affect the immune system of neonatal animals, leading to dysbiosis of the gut flora and drug resistance, among others [4,5,6]. Additionally, the overuse of antibiotics can lead to antibiotic residue in animal meat and food products, indirectly affecting the environment through animal excreta [7,8]. Faced with this controversial issue, some countries and regions have introduced restrictive policies on antibiotics added to feed, and alternatives to antibiotics have received increasing attention [9,10]. Probiotics such as *Lactobacillus*, *Bacillus subtilis*, and yeast are able to remodel the gut microbiota, alleviate ecological dysbiosis of the intestinal flora, attenuate intestinal inflammation [11,12,13], and improve immune, antioxidant, and production performance [14,15,16]; thus, fermented feed containing probiotics can be used to replace antibiotics in livestock feed.

Fermented feed is a product obtained by fermentation under appropriate conditions, using a single or complex probiotic and feed substrate. After fermentation, the probiotics used for fermentation dominate the feed, and their metabolites are able to lower the feed pH, reduce harmful microorganisms [17,18,19], inhibit the production of mycotoxins, eliminate allergens and antinutritional factors [20], and increase the crude protein content of the feed [21,22,23]. The large number of alcohols, ketones, esters, and acids produced in the fermentation process reportedly improve feed flavor [24]. Lu et al. concluded that fermented diets could promote the secretion of orexin and IGF-1 in the gastrointestinal tract and increase the number of IGF receptors to promote the growth of weaned piglets [25]. Previous research concluded that supplementation with fermented diets reduced the feed-to-weight ratio of piglets at 28–42 days of age and improved small intestinal barrier function and cecal microbial structure [26]. Xie et al. found that fermented soybean meal improved serum and muscle antioxidant capacity and meat quality in finishing pigs [27]. Other research concluded that fermented soybean meal had a positive effect on small intestine morphology in weaned piglets [28].

Feeding fermented diets could promote the expression of immune genes, secretion of cytokines and immunoglobulins, and change the structure of the colonic microbiota in growing finishing pigs [29]. Koo et al. concluded that feeding fermented barley to weaned piglets infected with *Escherichia coli* had similar effects to supplementation with an antimicrobial growth promoter [30]. The effect of *Salmonella enterica Serovar Typhimurium DT104* on intestinal infections was reduced in weaned piglets by feeding with *Lactobacillus*-fermented feed [31].

Huzhu Bamei (*Sus scrofa*), a unique pig breed farmed in the special climate of the northwest Qinghai–Tibet Plateau, has excellent livestock characteristics, including high adaptability, tolerance to hunger and cold, tolerance to roughage, stable genetic traits, and a strong reproductive ability [32]. It has been designated as a protected breed of livestock in China; however, purebred piglets present problems, including slow growth, respiratory disease, and post-weaning stress. We hypothesized that fermented feed could promote the healthy growth of Bamei piglets in the high-plateau environment. Therefore, this study investigated the effects of *L. plantarum*-fermented feed, *B. subtilis*-fermented feed, or a mixture of the two fermented feeds on the growth condition, immunity, and intestinal microbial flora. The findings of this study offer a valuable resource for the antibiotic-free rearing of Bamei piglets in the high plateau.

## 2. Materials and Methods

### 2.1. Characteristics of Bacterial Strains

The morphological, physiological, and biochemical properties of the two representative strains are listed in Table 1. Both strains were isolated from Bamei pig feces in Huzhu County, Qinghai, China [33]. Both were Gram-positive bacteria. *L. plantarum* QP28-1 grew well in 3.5% and 6.5% NaCl medium, at pH 4–10, and at 10–45 °C, but they grew weakly at 5 °C and 50 °C. *B. subtilis* QB8 grew well in 6.5% NaCl medium, at pH 5–10, and at 10–50 °C, but they did not grow well at 5 °C. The antibacterial activity of the colonies was assessed by measuring the diameter of the inhibition zones against pathogens using the double-layered agar plate method [33]. As shown in Table 1, the strains both presented strong antimicrobial activity against common pathogens.

### 2.2. Fermented Feed Preparation

Basic feed was produced by Shanxi Yangling First-feed Co., Ltd. (Shanxi, China). Based on the methods of Lu et al. and Guo et al. [25,34], the preparation of fermented feed was carried out as follows: A 1 L aqueous solution containing approximately 9 log colony-forming units /mL each of *L. plantarum* solution and *B. subtilis* solution was mixed with 1 kg of basal feed. After thorough mixing, it was packaged in plastic bags (2 kg/bag) with one-way valves. After sealing, the bags were kept indoors (30 ± 5 °C) for 5 days to obtain fermented feed. The feeds were prepared in the same manner each day, and a new bag of fermented feed was opened daily.

### 2.3. Analyses of the Fermented Feeds

All of the feed samples were dried to weigh the dry matter content based on the method reported by Liu et al. [35]. The methods used for the quantitation of the following constituents were in accordance with the Association of Official Agricultural Chemists (AOAC) Guidelines: crude protein (984.13), crude fat (2003.05), crude ash (942.05), neutral detergent fiber (2002.04), and acid detergent fiber (973.18).

Feed samples were mixed by vortexing with 1:10 deionized water for 20 min and then centrifuged at 3000× *g* for 8 min before the pH of the supernatant was determined.

### 2.4. Animal Experimental Design

Forty-eight Bamei piglets (25 days; 6.86 ± 0.97 kg) were equally divided into four groups, with each treatment comprising 3 replicates and 4 pigs in each replicate, resulting in a total of 12 animals per group: basal feed (CK group); diet containing 10% *L. plantarum*-fermented feed (L group); diet containing 10% *B. subtilis*-fermented feed (B group); and diet containing 5% *L. plantarum*-fermented feed and 5% *B. subtilis*-fermented feed (H group). The ingredients (%) of the diet are listed in Table 2. The diets contained neither antimicrobials nor growth promoters, and they met the nutrient requirements for swine (NRC, 2012). All Bamei piglets were acclimated to the basic diet and enclosure environment for 7 days in preparation for the experiment. The experiment was conducted at the Huzhu Bamei Pig Breeding Farm in Huzhu County, Qinghai Province, China. Piglets were housed in 3 × 3 m^2^ nursery beds equipped with slatted plastic floors, and the temperature was ~30 °C throughout the study period. The entire experiment lasted 28 days. Piglets were weighed at the beginning and end of the experiment, and feed consumption was recorded daily. Following the method described by Sun et al. [36], fecal samples with sticky and amorphous consistency were categorized as diarrhea, and the incidence was recorded at the same time every day for each group. The overall diarrhea rate was then calculated for the entire experiment period.

### 2.5. Collection of Fecal Samples

On days 0, 1, 3, 5, 7, 14, and 28 of the normal experiment, fresh fecal samples from the piglets were collected using sterile swabs and centrifuge tubes. Three samples were collected per replicate of each treatment group at each time point, resulting in a total of 252 samples. The samples were stored at - 80 °C.

### 2.6. Blood Sample Collection

Before morning feeding, blood samples were collected from all piglets through precaval venipuncture of the last day of the experiment. Samples were centrifuged at 4000× *g* for 10 min at 4 °C. The resulting serum was frozen at - 20 °C for further analysis.

The following serum parameters were determined using porcine ELISA kits (Dongge Boye, Beijing, China) targeting the following markers: interleukin (IL)-1β IL-6, tumor necrosis factor (TNF)-α, interferon (IFN)-γ, IgM, and Ig-G. Total protein (TP), lactate dehydrogenase (LDH), triglyceride (TG), urea nitrogen (UN), total cholesterol (TC), and alkaline phosphatase (AKP) were measured using commercial assay kits (Dongge Boye, Beijing, China). The steps of each assay were performed according to the manufacturer’s instructions.

### 2.7. 16S rRNA Sequencing

Total microbial DNA was isolated from excrements using a fecal DNA kit (DP712, Tiangen, Beijing, China), and the concentrations and purity of the DNA extraction were detected by agarose gel electrophoresis. The diluted DNA was used as a template for subsequent PCR experiments. Primers 341F (5′-CCTAYGGGRBGCASCAG-3′) and 806R (5′-GGACTACNNGGGTATCTAAT-3′) were used to amplify the V3-V4 region of the bacterial 16S rRNA gene. The PCR system consisted of 2 µM of primers and 10 ng of template DNA. PCR products were separated by 2% agarose gel electrophoresis and recovered using the Qiagen Gel Extraction Kit (Qiagen, Germany). The library was constructed and sequenced using the NovaSeq 6000 PE 250 (Novogene, Tianjin, China). The silver database (https://www.arb-silva.de/, accessed on 1 March 2023) was utilized to identify the detected microorganisms.

### 2.8. Statistical Analysis

A one-way ANOVA in IBM SPSS Statistics v. 27 (IBM-SPSS Inc., Chicago, IL, USA) was used to analyze the data on growth performance, immune indicators, biochemical indicators, and bacterial diversity. Tukey’s post hoc test was employed to analyze the significant differences between treatment groups. A significance level of *p* < 0.05 was considered statistically significant. Microbial sequencing data were processed using a cloud-based platform (https://magic.novogene.com, accessed on 21 May 2023). The QIIME2 (Version QIIME2-2020) was used to calculate observed OTUs, Shannon, Simpson, and Chao1 indices. Microbial relative abundance bar charts were constructed using the Origin software to compare the microbial proportions, and linear discriminant analysis was used to compare the characteristic microbial communities among the different treatment groups. Finally, Spearman’s correlation analysis was conducted to assess the relationship between the immune or biochemical indicators and bacterial abundance.

## 3. Results

### 3.1. Fermented Feed Characteristics

The chemical compositions of the different probiotic-fermented feeds are shown in Table 3. The *L. plantarum*-fermented and *B. subtilis*-fermented feeds showed the following changes: crude protein content was significantly increased (*p* < 0.05); ash, dry matter, crude fat, and acid detergent fiber tended to be decreased; neutral detergent fiber was significantly decreased (*p* < 0.05); and the pH values were significantly lower (*p* < 0.05).

### 3.2. Growth Performance and Diarrhea Rate of the Piglets

Based on the results of the four experimental groups, Figure 1 shows the growth performances of the Bamei piglets. There was no difference in initial body weight between the groups, whereas the mixed-fermented-feed-supplemented piglets had significant increases in ADFI and ADG compared to that of the control (CK) group (*p* < 0.05). The feed/gain ratios and diarrhea rate in all three of the fermented feed groups showed a decreasing trend compared to that of the CK group (*p* > 0.05).

### 3.3. Changes in Serum Immune Factor Concentrations of the Piglets

As shown in Figure 2A,B, compared with the CK group, the values of TNF-α and IL-1β in the Bamei piglets fed with *L. plantarum*-fermented feed (L) or *B. subtilis*-fermented feed (B) were significantly lower (*p* < 0.01), the IL-6 concentration was significantly lower in the three fermented feed groups (*p* < 0.05, Figure 2C), and all three of the fermented feed groups showed significantly elevated serum concentrations of IgM, IgG, and IFN-γ (*p* < 0.05, Figure 2D–F). Furthermore, the group of piglets fed a mixture of *L. plantarum*- and *B. subtilis*-fermented feed (H) exhibited higher serum levels of IgM, IgG, and IFN-γ than that of the groups fed with either of the single-fermented feeds (*p* < 0.05).

### 3.4. Changes in Serum Biochemical Factor Concentrations of the Piglets

Serum TP concentrations were significantly elevated (*p* < 0.05, Figure 3A), but TC and TG concentrations were significantly reduced (*p* < 0.05, Figure 3C,D) in the mixed H group compared to the CK group. All three of the fermented feeds decreased the level of serum UN compared with that of the control feed (Figure 3B). As shown in Figure 3E,F, there was no significant change in AKP or LDH concentrations among the four groups (*p* > 0.05).

### 3.5. Changes in the Intestinal Microbiota of the Piglets

The α-diversity of the observed operational taxonomic unit (OTU), Chao1, Simpson and Shannon indexes of the gut microbiota are shown in Figure 4. On day 7, the bacterial OTU index in the mixed-fermented feed group H was much higher than that of the other feed groups (*p* < 0.01). The Chao1 index was significantly higher in the L group on day 5 (*p* < 0.01) and in the H group on day 7 (*p* < 0.05) compared to that of the CK group. The Simpson index of the fecal bacterial samples was significantly higher in the H group compared to the L group on day 7 (*p* < 0.05). On day 7, the Shannon index was significantly higher in the H group than in the L and B groups (*p* < 0.01). On day 28, the bacterial OTU, Chao1, Simpson and Shannon indices of the piglets in the three fermented feed groups were higher than in the control piglets.

The relative abundances of the predominant bacteria in the Bamei piglets at the phylum level are shown in Figure 5. According to the sequencing analysis results, Firmicutes, Bacteroidota, Euryarchaeota, Spirochaetota, and Proteobacteria had the highest relative abundances in all four groups, accounting for 93.69–99.89% of the total bacteria. On day 7, the Firmicutes in the fermented feed groups began to increase; on day 28, they exceeded that of the CK group, with rates of 60.53%, 70.60%, 68.82%, and 68.19% in the CK, L, B, and H groups, respectively. Bacteroidetes were more abundant in the fermented feed groups than in the CK group on day 14, with rates of 13.87%, 19.63%, 20.62%, and 25.15% in the CK, L, B, and H groups, respectively. In group B piglets, the relative abundance of Bacteroidota gradually increased with feeding, from 20.8% on day 0 to its highest level of 35.50% on day 7, remaining stable thereafter. Proteobacteria were less abundant in the fermented feed groups than in the CK group on day 14, with rates of 3.15%, 0.91%, 1.00%, and 0.84% in the CK, L, B, and H groups, respectively. The relative abundances of Proteobacteria dropped to the lowest levels on day 5 in the L and H groups (0.04% and 0.02%, respectively), and on day 7 in the B group (0.32%).

*Lactobacillus*, *Muribaculaceae*, *Methanobrevibacter*, *Clostridium_sensu_stricto_1*, *Prevotella*, and *Parabacteroides* dominated in terms of relative abundance in all four groups at the genus level (Figure 6). The relative abundances of *Lactobacillus* were higher in the L, B, and H groups than in the CK group on days 7, 14, and 28. On day 28, the relative abundances of *Lactobacillus* were 2.24%, 5.55%, 5.72%, and 4.84% in the CK, L, B, and H groups, respectively. On day 14, *Muribaculaceae* in the L and B groups were more than the CK group (6.52%), with rates of 8.49% and 9.58%, respectively. The relative abundances of *Methanobrevibacter* in the fermented feed groups were lower than that in the CK group on days 14 and 28, with rates of 8.69%, 5.06%, 2.74%, and 4.43% in the CK, L, B, and H groups, respectively, on day 28. On day 14, *Clostridium_sensu_stricto_1* was reduced in the fermented feed groups, with rates of 16.24%, 8.16%, 14.47%, and 13.47% in the CK, L, B, and H groups, respectively. The relative abundances of *Prevotella* were increased in the fermented feed groups on days 14 and 28, with rates of 0.27%, 1.31%, 2.02%, and 1.51% in the CK, L, B, and H groups, respectively, on day 28.

LEfSe analysis of the bacterial biomarkers at 28 days is shown in Figure 7. Compared with the CK group, the L group had enrichment of the following genera: *Rikenellaceae_RC9_gut_group*, *Alloprevotella*, *Holdemanella*, *Fournierella*, *Coprococcus*, *Prevotellaceae_NK3B31_group*, and *Lachnospiraceae_NK4A136_group* (Figure 7A). the B group had enrichment of the following genera: *Eubacterium_hallii_group*, *Clostridia_UCG_014*, *Dorea*, *Coprococcus*, *Lachnospiraceae_NK4A136_group*, *Eubacterium_oxidoreducens_group*, *UCG_005*, *Rikenellaceae_RC9_gut_group*, and *Prevotella*. The species *Lactobacillus_pontis*, *Lactobacillus_delbrueckii* (Figure 7B), *Prevotellaceae_NK3B31_group*, *Prevotella* and *Alloprevotella* were more abundant in group H (Figure 7C).

### 3.6. Correlations between the Serum Immune and Biochemical Factors and the Intestinal Microbiota

The correlation analysis between the microbial abundance in the fecal samples of the four treatment groups on day 28 and the serum biochemical and immune factor levels are shown in Figure 8. At the phylum level, IL-6 was negatively correlated with Firmicutes (*p* < 0.05) and positively correlated with Bacteroidota (Figure 8A,C). TP was positively correlated with Firmicutes (*p* < 0.05). AKP was negatively correlated with Fusobacteriota (*p* < 0.05). Figure 8B,D illustrates that, at the genus level, *Rikenellaceae_RC9_gut_group* was positively correlated with IgG (*p* < 0.01) and negatively correlated with IL-1β (*p* < 0.01).

## 4. Discussion

The nutritional properties of feed can be improved by fermentation [37,38]. In our study, the crude protein values of feed were improved by fermentation with *L. plantarum* and *B. subtilis*. Reductions in crude ash, fiber, and dry matter were similar to previous studies [17,22,39]. Increases in crude protein and decreases in fiber may be caused by the increases in the protease and cellulase activities of probiotics during the fermentation process. Low pH is an important quality indicator of fermented feed, helping to inhibit the growth of undesirable microorganisms and thus prevent spoilage. Our study found that the pH of the *L. plantarum*- and *B. subtilis*-fermented feeds decreased to 4.44 and 4.71, respectively, possibly caused by the proliferation of lactic acid bacteria in fermentation [40,41].

Fermented feed preparations generally improve the growth of pigs [42,43]. In our study, the ADFI was higher in the three fermented feed groups compared with that in the CK group, which may have been related to the elimination of allergens and antinutritional factors and the improved palatability resulting from the fermentation process [20,44]. The large number of alcohols, ketones, esters, and acids produced in the fermentation process improved feed flavor and elevated serum levels of orexins [24,25]. All three fermented feed groups showed improvements in ADG content and F/G ratio, which is consistent with previous reports [45]. Of the three groups, the mixed fermented feed group had the most favorable ADFI, ADG, and F/G ratios, suggesting that co-feeding *L. plantarum*- and *B. subtilis*-fermented feed may be more suitable than single-fermented feed for Bamei piglets. Probiotic-fermented feed is generally effective in improving diarrhea in pigs. Wang et al. found that adding fermented soybean meal to the diet alleviated diarrhea caused by *enterotoxigenic Escherichia coli* K88 in piglets [46], and Lin et al. discovered that feeding piglets with *Bacillus licheniformis*-fermented feed had a similar effect to antibiotic treatment in relieving diarrhea [47]. Our results also indicate that fermented feed can reduce the diarrhea rate in Bamei piglets.

Cytokines and immunoglobulins are often used as indicators of the body’s immune status, playing important roles in immune regulation and defense [48,49]. Immunoglobulins, which are mainly found in serum, recognize pathogens and prevent pathogenic infestations and potentially harmful microbial aggressions [50,51,52]. Studies in weaned piglets have shown that an *L. plantarum* LQ80-fermented fluid diet reduced TNF-α secretion and increased the total serum IgM and IgG levels compared with an unfermented fluid diet [53]. Cheng et al. showed that feeding soybean meal fermented by complex probiotics could increase the serum immunoglobulin concentration to improve immunity in weaned piglets [28,54]. Cytokines also play an important role in cell-mediated immune responses. Wang et al. found that combined feeding of *Lactobacillus* fermentum and *Lactococcus lactis* reduced the levels of IL-6, IL-1β, and TNF-α in weaned piglets, helping to minimize inflammation in piglets. [55]. Li et al. found that probiotic bacteria isolated from yaks reduced TNF-α and IL-6 in mice while increasing serum immunoglobulins [56]. Similarly, in our study on Bamei pigs, dietary supplementation with three different fermented feeds improved immunity, as mainly reflected by the significant increases in serum IgM and IgG levels, significant reductions in serum IL-6 levels, and reductions in TNF-α and IL-1β compared to the basic feed. These effects were most obvious in the mixed-fermented feed group. Among immune-related factors, IFN-γ levels were relatively higher in the three fermented feed groups. Some studies have shown that when the intestinal mucosa is stimulated by probiotics, the increase in Toll-like receptor expression in intestinal mucosal cells leads to increased release of cytokines such as IFN-γ [57]. Preweaning calves fed with *B. subtilis* natto were found to secrete more IFN-γ and IgG than control calves [58]. Elevation in cytokines such as IFN-γ was also found in hens fed with fermented feed [59]. The results of these reports are consistent with ours.

AKP plays important roles in calcium and phosphorus uptake, protein synthesis, and bone calcification during animal growth and development [60]. TP and UN levels respond to protein synthesis and metabolism during piglet growth. Because UN is a degradation product of protein, the elevation in serum TP and reduction in serum UN levels during piglet growth in this study indicated that protein synthesis and uptake were improved [61]. Zhu et al. reported that piglets given with 10% fermented soybean meal had better growth improvement and relatively high serum AKP and TP levels compared to the control and 5% fermented soybean meal groups. Furthermore, serum UN was reduced in the experimental group relative to the blank group [28]. Supplementing chow with probiotics isolated from yak improved the growth performance of mice, it also raised AKP and TP levels [56]. High serum cholesterol levels are associated with cardiovascular diseases, and some studies have demonstrated the cholesterol-lowering potency of probiotics and fermented products [62,63]. Studies in broiler chickens reported that cholesterol and TG were reduced after feeding with fermented rapeseed meal [64]. Similarly, in this study, TP and AKP levels were increased, and UN, TC, and TG levels were reduced in the groups of piglets supplemented with fermented feed.

Intestinal microorganisms perform important roles in the growth and immunity of piglets. We found that 28 days on any of the three fermented-feed-supplemented diets led to improvements in OTU, Chao1, Simpson, and Shannon indices, indicating that fermented feed supplementation can improve the diversity of the gut microbiota. The fecal microbiota of pigs reportedly mainly comprises five phyla: Firmicutes, Bacteroidetes, Proteobacteria, Actinobacteria, and Spirochaetes [65]. The intestinal microflora of the Bamei piglets in our study was dominated by Firmicutes, Bacteroidota, Euryarchaeota, Spirochaetota, and Proteobacteria. Differences in gut microbiota may be influenced by the breed, age, diet, and living environments of the pigs [66,67], and such differences have been reported between populations living at high altitudes and terrestrial areas [68].

In our study, the fermented feed groups developed an increase in Firmicutes and Bacteroidetes and a decrease in Proteobacteria. In previous reports, Firmicutes and Bacteroidetes were the two most abundant phyla in pig feces [69]. Intestinal fermentation of fiber by Firmicutes and Bacteroidetes produces partial fatty acids, which act on enteroendocrine cells to affect host metabolism [70]. Enhanced Firmicutes and Bacteroidetes have been reported to increase the ability of animals to acquire energy [71,72]. Phylum Proteobacteria include many pathogenic bacteria, and an increase in the phylum is considered a potential feature of disease risk [73,74]. Lactic acid bacteria have the ability to resist pathogens, and they are anti-inflammatory, antioxidant, and cholesterol-lowering [75,76], having positive impacts on the growth and immunity of animals [77,78]. In this study, the abundances of *Lactobacillus* and *Muribaculaceae* were increased and those of *Methanobrevibacter* and *Clostridium* were reduced after fermented feed supplementation. *Muribaculaceae* belongs to the phylum Bacteroidota and is reportedly associated with the degradation of various complex carbohydrates, inhibition of harmful bacteria and oxidative stress, improvement of intestinal inflammation, and lowering of blood lipids [79,80,81]. *Methanobrevibacter* has been categorized into the phylum Euryarchaeota, and the decrease in *Methanobrevibacter* abundance in our fermented feed groups may have led to reduced methane production and lowered energy loss. It has been reported that *Clostridium_sensu_stricto* may be associated with intestinal inflammation [82,83,84]. In this study, *Prevotellaceae_NK3B31_group* was negatively correlated with TNF-α and UN and positively correlated with IgG, AKP, and TP, while *Rikenellaceae_RC9* was positively correlated with IgG and negatively correlated with IL-1β. *Prevotellaceae_NK3B31_group* belongs to the genus *Prevotella*, which predominates in the gastrointestinal tracts of nursery piglets, helping to produce intramuscular fats and store hepatic glycogen. *Prevotella* also reduces inflammation by decreasing intestinal permeability [85]. The family *Rikenellaceae* belongs to the phylum Mycobacterium, which produces propionate and acetate as fermentation products [86]. It has been reported that *Rikenellaceae* is significantly associated with weight gain in weaned piglets [87]. Therefore, the increased abundances of the *Prevotellaceae* and *Rikenellaceae* families in the intestinal tract of Bamei piglets on fermented-feed-supplemented diets may have contributed to their improved growth performance.

Probiotic-fermented feed can improve the growth of Bamei piglets by modulating the cytokine levels and the composition of the gut microbiome. However, future research needs to comprehensively consider the optimal parameters and economic benefits of feed fermentation, the impact of fermented feed on the intestinal development of Bamei piglets, and the effects of different caloric intakes and water consumption during episodes of diarrhea on their growth status.

## 5. Conclusions

Fermentation by *L. plantarum* QP28-1 or *B. subtilis* QB8 improved the nutritional value of basic pig feed. The supplementation of basic control feed with fermented feed improved the growth performance, immune status, and intestinal microbiota of Bamei piglets. Furthermore, supplementation with a mixture of *L. plantarum* QP28-1- and *B. subtilis* QB8-fermented feed was superior to that of either of the single-fermented feeds. This study provides a new approach for using fermented feed as an in-feed antibiotic substitute in the raising of Bamei piglets in high-altitude environments. Further research is needed on the long-term effects of fermented feed on the immune system of Bamei pigs, as well as how to maximize the use of fermented feed to improve intestinal health and microbial diversity.

## Figures and Tables

**Figure 1 animals-14-00647-f001:**
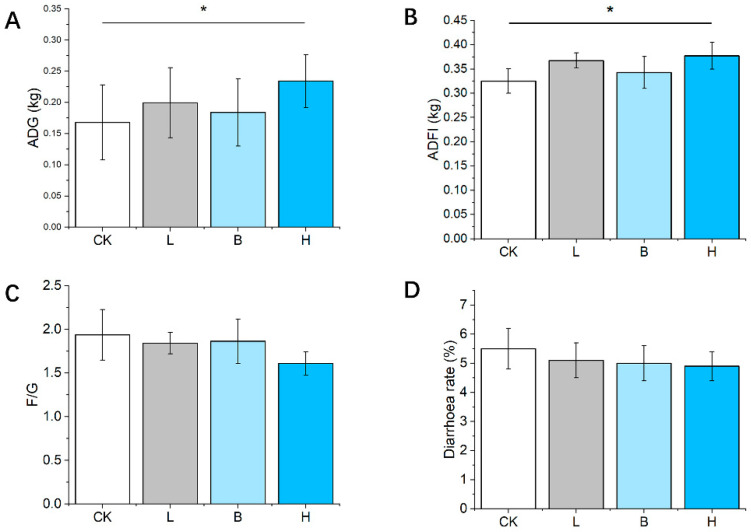
The growth performance and diarrhea rate of Bamei piglets. (**A**–**D**) The ADG, ADFI, F/G, and diarrhea rate in the different groups. Abbreviations: ADFI = average daily feed intake; ADG = average daily weight gain; F/G = ratio of feed intake to gain; CK = basic diet control; L = basic diet with 10% *L. plantarum*-fermented feed; B = basic diet with 10% *B. subtilis*-fermented feed; H = basic diet with 5% *L. plantarum*-fermented feed and 5% *B. subtilis*-fermented feed. All values are means ± SEM (*n* = 12 per group). * *p* < 0.05.

**Figure 2 animals-14-00647-f002:**
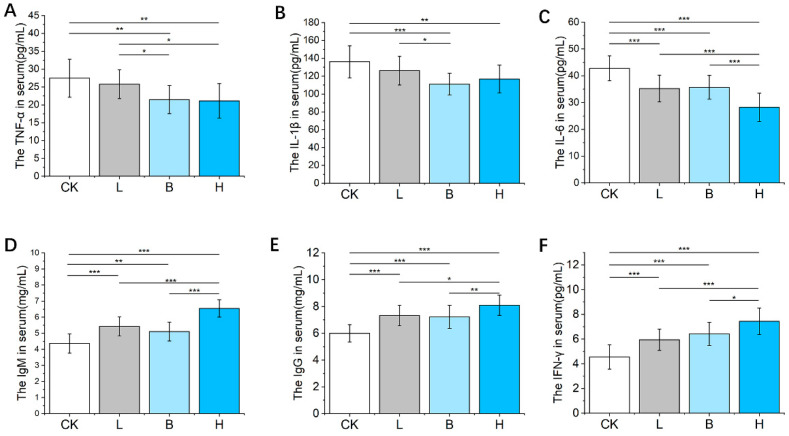
The effects of fermented feed on the serum immune factors in Bamei piglets. (**A**–**F**) The serum TNF-α, IL-1β, IL-6, IgM, IgG, and IFN-γ levels in the different groups. Abbreviations: Ig = immunoglobulin; IL = interleukin; TNF = tumor necrosis factor; IFN, interferon; CK = basic diet control; L = basic diet with 10% *L. plantarum*-fermented feed; B = basic diet with 10% *B. subtilis*-fermented feed; H = basic diet with 5% *L. plantarum*-fermented feed and 5% *B. subtilis*-fermented feed. All values are means ± SEM (*n* = 12 per group). * *p* < 0.05, ** 0 *p* < 0.01, *** *p* < 0.001.

**Figure 3 animals-14-00647-f003:**
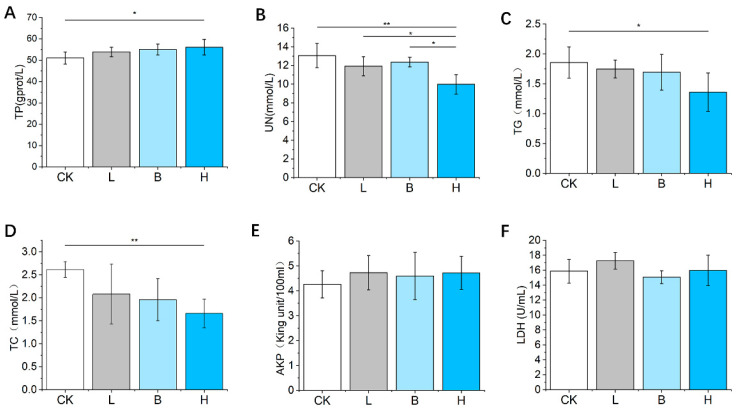
The effects of fermented feed on the serum biochemical concentrations in Bamei piglets. (**A**–**F**) The serum TP, UN, TG, TC, AKP and LDH levels in the different groups. Abbreviations: TP = total protein; UN = urea nitrogen; TG = triglyceride; TC = total cholesterol; AKP = alkaline phosphatase; LDH = lactate dehydrogenase; CK = basic diet control; L = basic diet with 10% *L. plantarum*-fermented feed; B = basic diet with 10% *B. subtilis*-fermented feed; H = basic diet with 5% *L. plantarum*-fermented feed and 5% *B. subtilis*-fermented feed. All values are means ± SEM (*n* = 12 per group). * *p* < 0.05, ** *p* < 0.01.

**Figure 4 animals-14-00647-f004:**
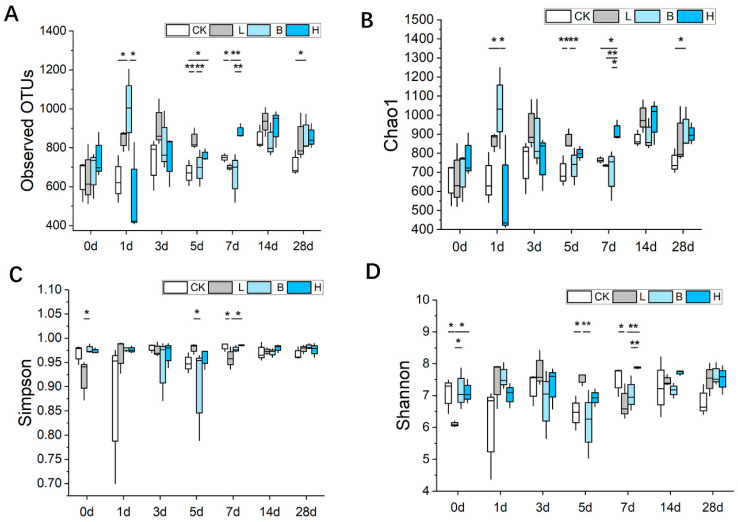
The effects of fermented feed on the microbiota α-diversity indices, including (**A**) the observed OTUs, (**B**) Chao1, (**C**) Simpson, and (**D**) Shannon indices of Bamei piglet feces. Abbreviations: CK = basic diet control; L = basic diet with 10% *L. plantarum*-fermented feed; B = basic diet with 10% *B. subtilis*-fermented feed; H = basic diet with 5% *L. plantarum*-fermented feed and 5% *B. subtilis*-fermented feed. All values are means ± SEM (*n* = 12 per group). * *p* < 0.05, ** *p* < 0.01.

**Figure 5 animals-14-00647-f005:**
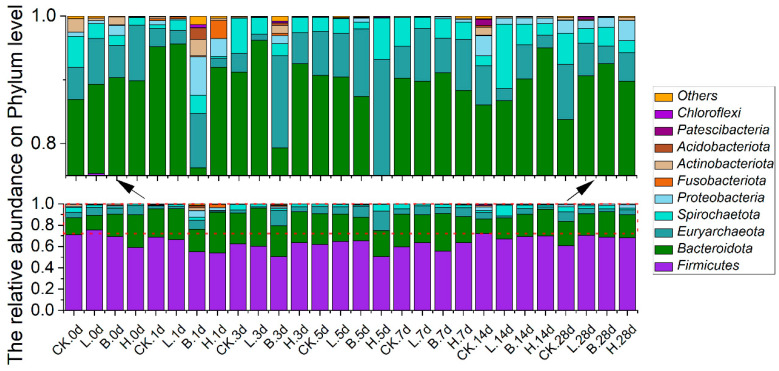
The relative abundances of predominant bacteria at the phylum level in different growth periods of Bamei piglets. Abbreviations: CK = basic diet control; L = basic diet with 10% *L. plantarum*-fermented feed; B = basic diet with 10% *B. subtilis*-fermented feed; H = basic diet with 5% *L. plantarum*-fermented feed and 5% *B. subtilis*-fermented feed.

**Figure 6 animals-14-00647-f006:**
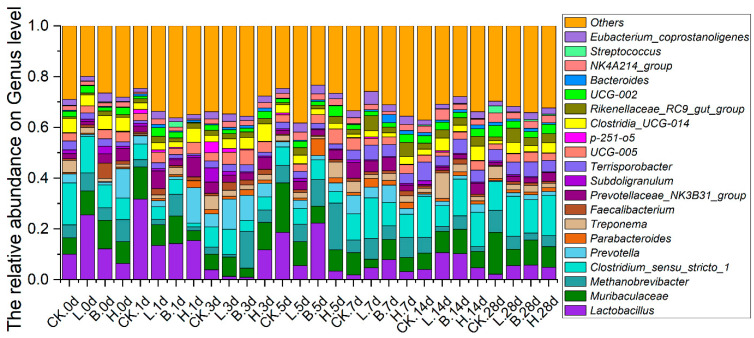
The relative abundance of predominant bacteria at the genus level in the different growth periods of Bamei piglets. Abbreviations: CK = basic diet control; L = basic diet with 10% *L. plantarum*-fermented feed; B = basic diet with 10% *B. subtilis*-fermented feed; H = basic diet with 5% *L. plantarum*-fermented feed and 5% *B. subtilis*-fermented feed.

**Figure 7 animals-14-00647-f007:**
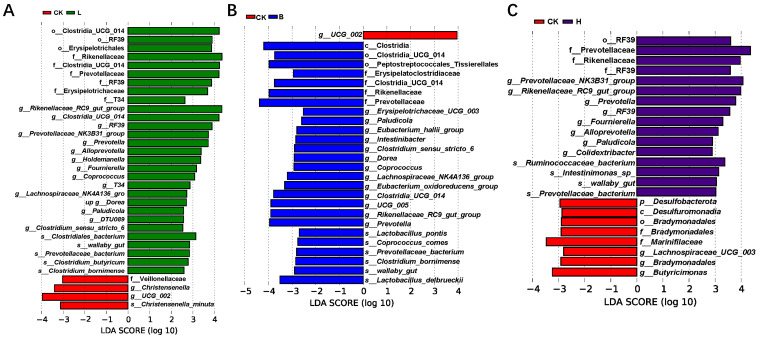
Linear discriminant analysis (LDA) combined with the effect size for microbiota enrichment in the different groups of Bamei piglets at 28 days (LDA Score > 3, *p* < 0.05). (**A**) The CK and L groups, (**B**) CK and B groups, and (**C**) CK and H groups. Abbreviations: CK = basic diet control; L = basic diet with 10% *L. plantarum*-fermented feed; B = basic diet with 10% *B. subtilis*-fermented feed; H = basic diet with 5% *L. plantarum*-fermented feed and 5% *B. subtilis*-fermented feed.

**Figure 8 animals-14-00647-f008:**
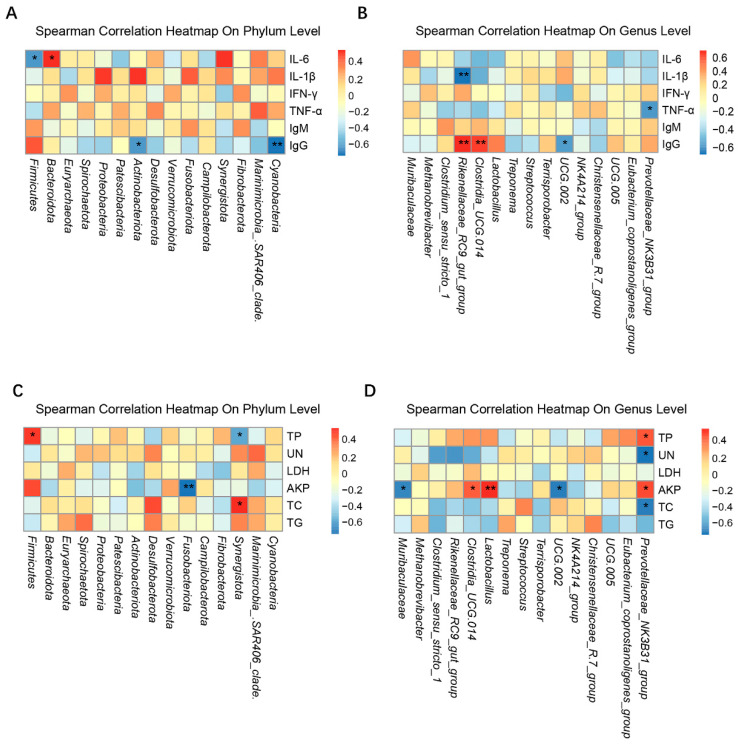
Correlations between the serum immune and biochemical factors and the intestinal microbiota of Bamei piglets. (**A**,**C**) The phylum level and (**B**,**D**) genus level. The blue and red grids indicate negative or positive correlation (* *p* < 0.05, ** *p* < 0.01), respectively. Abbreviations: Ig = immunoglobulin; IL = interleukin; TNF = tumor necrosis factor; IFN, interferon; TP = total protein; LDH = lactate dehydrogenase; UN = urea nitrogen; TC = total cholesterol; AKP = alkaline phosphatase; TG = triglyceride.

**Table 1 animals-14-00647-t001:** Morphological, physiological and biochemical properties of *L. plantarum* QP28-1 and *B. subtilis* QB8.

Character	QP28-1	QB8
Species	*Lactobacillus plantarum*	*Bacillus subtilis*
Collecting location	Huzhu County, Qinghai, China	
Sample sources	Feces of Bamei pig	
Shape	rod	rod
Gram stain	+	+
Growth in NaCl		
3.5	++	−
6.5	++	++
Growth performance at Temperature (°C)		
5	w	−
10	+	+
30	++	++
45	+	++
50	w	+
Growth performance at pH		
3	−	−
4	+	−
5	++	+
8	++	+++
9	++	++
10	+	+
Antibacterial activity		
*Escherichia coli ATCC 30105*	++	+
*Micrococcus luteus ATCC 4698*	+++	++
*Staphvlococcus aureus ATCC 29213*	++	++
*Pseudomonas aeruginosa ATCC 27853*	++	−
*Listeria monocytogenes BAA*	++++	++++
*Salmonella enterica ATCC 43971*	++	++

Note: For temperature, NaCl, and pH tolerance: −, OD600 = 0, no growth; w, 0 < OD600 < 0.2, growth weak; +, 0.2 < OD600 < 0.6, well growth; ++, grow very well 0.6 < OD600. For antibacterial activity, the inhibition zone contains the external diameter of the cup (10 mm). The diameter of the inhibition zone: −, no inhibition; +, 10–15 mm; ++, 15–20 mm; +++, 20–25 mm; ++++, more than 25 mm.

**Table 2 animals-14-00647-t002:** The nutrient compositions of the basal diet (air-dry basis).

Ingredients	Contents (%)	Nutritional Indicators ^2^	
Corn	60.00	Digestible energy (MJ/kg)	14.78
Soybean meal	26.00	DM (%)	89.16
Fish meal	2.00	CP (%)	19.45
Wheat bran	3.00	Crude fiber (%)	3.12
Whey powder	5.00	Crude fat (%)	2.15
Premix ^1^	4.00	Crude ash (%)	5.18
Total	100	ADF (%)	4.15
		NDF (%)	13.45
		NaCl (%)	0.50
		Calcium (%)	1.02
		Total Phosphorus (%)	0.58

Note: ^1^ Premix supplied per kilogram of meal: vitamin E, 15 mg; vitamin A, 12,000 IU; vitamin B6, 5 mg; vitamin B12, 0.2 mg; vitamin D3, 500 IU; Mn, 45 mg; Zn 120 mg; Fe, 220 mg; Cu, 95 mg; I 0.45 mg; Se, 0.5 mg; folic acid, 5 mg; biotin, 0.3 mg; and pantothenic acid, 20 mg. ^2^ The analyzed compositions are the calculated/measured values.

**Table 3 animals-14-00647-t003:** Chemical composition of basic feed before and after fermentation.

Items	Prior to Fermentation	After Fermentation		SEM	*p*-Value
		*L. plantarum* QP28-1 Fermented Feed	*B. subtilis* QB-8 Fermented Feed		
CP (%)	19.45 ^B^	22.63 ^A^	22.80 ^A^	0.13	0.017
Crude fat (%)	2.15	1.86	1.81	0.05	0.055
Crude ash (%)	5.18	4.91	4.85	0.05	0.052
ADF (%)	4.15	4.05	4.00	0.02	0.071
NDF (%)	13.35 ^A^	12.17 ^B^	11.5 ^B^	0.03	0.047
DM (%)	89.16	87.11	87.35	1.24	0.061
pH	6.24 ^A^	4.44 ^B^	4.71 ^B^	0.15	0.009

Note: ^AB^ The different letters on one line denote a significant difference (*p* < 0.05).

## Data Availability

Data were not uploaded in the repository. Relevant data can be obtained from the authors.

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
