# Peer review of "Effects of Probiotic-Fermented Feed on the Growth Profile, Immune Functions, and Intestinal Microbiota of Bamei Piglets"

_animals, 2024, doi:10.3390/ani14040647_

Round 1

Reviewer 1 Report

Comments and Suggestions for Authors

Dear Authors,

the work presented is valuable. The introduction is well written and does not raise any major objections. You can add here information about the fermentation process itself and its impact on the feed. The material and methods should be described in more detail. The results presented in the table require reference in the methodology. How were the antimicrobial properties against pathogens assessed? Please provide fermentation conditions and feed composition before and after fermentation. Was consent obtained from the ethical committee for animal experiments? Metagenomic analyzes of the composition of microorganisms are very valuable. The discussion is well written. 

Reviewer 2 Report

Comments and Suggestions for Authors

1.       Bacterial names should be in italics. Please check and correct throughout the manuscript.

2.       The process of fermented feed preparation is not clear. References are missing. How much bacteria is mixed in a 20kg bag?

3.       The authors did 5 days incubation of feed with bacterial strains. How did they decide on this time point? Primarily it should be tested for different time points or should provide some references.

4.       It will be better to show the weight increase and feed/weight ratio in graphical format.

5.       Conclusions should be extended including future perspectives and applications.

6.       There are many associated limitations to the current study. Please include a section describing the limitations of the current study. 

Reviewer 3 Report

Comments and Suggestions for Authors

Dear Authors,

thank you for providing this manuscript. I have several comments and questions. Please find them per line below:

Line 20-23: Could you please add the group names in brackets?

Line 23-25: please use the group codes in brackets here too.

Line 44: as antibiotics not only affects the immune system of neonatal animals, please add among others or rephrase

Line 71: please put the whole name of the microorganism to italic letters

Line 92: Please ad (see Table 1)

Line 103/104/ 292 and following/408: please put the names of the microorganisms to italic letters

Did you ferment the food for each group for each day the same way (so 5 day fermentation) or did you use one bag of fermented food as long as it was finished? How long did it take to empty one bag? Did you analyse the fermentation at opening or at finishing one bag?

Line 117: Each treatment should have 3 replicates of 4 pigs each resulting in 12 animals per group. Could you please clarify this in your text?

Line 135: The collection of fecal samples is not clear for me. Please explain, how did you get a total of 48 samples. 3 replicates of 4 groups (if the stool of one group is pooled) with 7 timepoints of sample taking should result in 84 samples as I understood this right. If the stoole is not pooled, there should be 336 samples. Please clarify!

Line 141: please ad (if this is correct): of the last day of the experiment

                How often did you take blood? Please state!

Line 151: please state exactly what you defined as fecal samples

Line 159: please give the database (silver, greengene…) used for determining the found microorganisms

Statistical analysis: Which post test did you use? A Wilcoxon test is only allowed for two experimental groups. As you have four groups this does not fit. Please recalculate the statistics completely with appropriate methods! Please list also the statistics you did for microbiome analysis here.

You found out, that the different food mixtures were of significantly different composition. And then you fed the animals with significantly different amounts of food daily. How can you fix that the differences found in the piglets concerning weight is not because of the different amount of food and so the different calory intake? There is a difference in food intake of up to 1680g over your experimental phase! What did you take as equalizing component between the groups to make a direct comparison possible?

Did you check for diarrhea and protocol this for the study phase as it its absence is a very strong sign for wellbeeing in weaned piglets? Also the different calory intake and loose of water when suffering from diarrhea needs to be taken into account in this study. Please insert this or place it in the study limitations.

Figure 4: Is it possible to enlarge the part between 85% and 100% to make your results visible?

Figure 6: Please sort the results for the different levels (genus, species, family…) to make the comparison easier.

The values of which groups did you use for correlation analysis?

Line 355: Please put a . before Furthermore

Line 371 please add be between may and influenced

Line 381: please introduce the abbreviation LAB

Please ad study limitations concerning diarrhea, different amounds of food/calorie intake and so on.

kind regards

Comments on the Quality of English Language

Dear Authors,

the manuscript is written in good English. It is easy to follow and good to understand. I have found some small matters. Please find them listed per line above.

kind regards

Round 2

Reviewer 2 Report

Comments and Suggestions for Authors

The authors successfully responded to the reviewers comments and updated the manuscript as well. 

Reviewer 3 Report

Comments and Suggestions for Authors

Dear Autors,

thank you for taking my suggestions into account! The quality of your manuscript has improved a lot and everything is now clearly described.

I have only one point which I would like to ask you to correct:

The figures 5,6,7 seem to lay above other figures. I can only see the border which implicates something is underneath. One figure legend has also moved in between the text leaving its figure (6). Maybe this happened while converting to .pdf. Could you please check?

Thank you!

kind regards